# Impact of Breathing Phase, Liver Segment, and Prandial State on Ultrasound Shear Wave Speed, Shear Wave Dispersion, and Attenuation Imaging of the Liver in Healthy Volunteers

**DOI:** 10.3390/diagnostics13050989

**Published:** 2023-03-05

**Authors:** Catherine Paverd, Sivert Kupfer, Iara Nascimento Kirchner, Sherin Nambiar, Alexander Martin, Naiara Korta Martiartu, Thomas Frauenfelder, Marga B. Rominger, Lisa Ruby

**Affiliations:** 1Institute of Diagnostic and Interventional Radiology, University Hospital Zurich, Rämistrasse 100, 8091 Zurich, Switzerland; 2Institute of Applied Physics, University of Bern, Sidlerstrasse 5, 3012 Bern, Switzerland

**Keywords:** ultrasound liver assessment, shear wave elastography, shear wave speed, shear wave dispersion, attenuation imaging

## Abstract

Objectives: Measurement location and patient state can impact noninvasive liver assessment and change clinical staging in ultrasound examinations. Research into differences exists for Shear Wave Speed (SWS) and Attenuation Imaging (ATI), but not for Shear Wave Dispersion (SWD). The aim of this study is to assess the effect of breathing phase, liver lobe, and prandial state on SWS, SWD, and ATI ultrasound measurements. Methods: Two experienced examiners performed SWS, SWD, and ATI measurements in 20 healthy volunteers using a Canon Aplio i800 system. Measurements were taken in the recommended condition (right lobe, following expiration, fasting state), as well as (a) following inspiration, (b) in the left lobe, and (c) in a nonfasting state. Results: SWS and SWD measurements were strongly correlated (r = 0.805, *p* < 0.001). Mean SWS was 1.34 ± 0.13 m/s in the recommended measurement position and did not change significantly under any condition. Mean SWD was 10.81 ± 2.05 m/s/kHz in the standard condition and significantly increased to 12.18 ± 1.41 m/s/kHz in the left lobe. Individual SWD measurements in the left lobe also had the highest average coefficient of variation (19.68%). No significant differences were found for ATI. Conclusion: Breathing and prandial state did not significantly affect SWS, SWD, and ATI values. SWS and SWD measurements were strongly correlated. SWD measurements in the left lobe showed a higher individual measurement variability. Interobserver agreement was moderate to good.

## 1. Introduction

Liver diseases, including alcohol-associated liver disease, nonalcoholic fatty liver disease, viral hepatitis, drug-induced liver injury, and other autoimmune and metabolic disorders account for over two million deaths annually [1]. Liver diseases cause pathological structural changes, including fibrosis (increased collagen and extracellular matrix deposition), steatosis (a build-up of fats), and inflammation (an infiltration of inflammatory cells or upregulation of inflammatory mediators) [2,3]. Noninvasive liver assessment of these characteristics is crucial to enable clinicians to optimize treatment. Currently, fibrosis and steatosis are routinely assessed by two quantitative ultrasound metrics: stiffness and attenuation [4].

Stiffness is calculated from Shear Wave Speed (SWS) measurements obtained during Shear Wave Elastography (SWE) procedures. In SWE, shear waves are generated as a result of tissue deformation. The deformation is created by mechanical waves arising either from a physical vibrating object against the skin (Transient Elastography, TE), or from the Acoustic Radiation Force Impulse (ARFI) generated from a focused ultrasound beam (a ‘push pulse’) [4]. The speed at which shear waves travel is directly related to the tissue stiffness [5] and, hence, to fibrosis [6,7,8,9]. Attenuation measurements are calculated from the decrease in signal amplitude with depth and evaluated in a region of interest defined by the machine or the operator. For example, using the Controlled Attenuation Parameter (CAP) of the FibroScan TE device (Fibroscan, Echosens, Paris, France), which is calculated from a decrease in signal strength over a machine-defined region within the liver, a single attenuation value is displayed in units of dB/m. With other vendors (for example, Canon, General Electric, Samsung, or Siemens), a 2D Attenuation Imaging (ATI) map can be obtained, and the final region of interest within the ATI map can be selected by the operator to obtain a single ATI value displayed in dB/cm/MHz [10]. Both CAP [11] and ATI [12,13] can be used to assess steatosis.

Recently, Canon Medical Systems Corporation (Otawara, Japan) introduced a novel metric called Shear Wave Dispersion (SWD), which in some studies to date appears to be an indicator of liver inflammation in humans [14,15]. SWD aims to assess the physical property of viscosity. The liver is usually modelled as a purely elastic medium for stiffness measurements, but contains both viscous and elastic components. In viscoelastic media, SWS increases monotonically with frequency [6]. Thus, a shear wave in the liver will undergo dispersion, and the gradient of the change in SWS with respect to frequency (the ‘dispersion slope’) is directly related to viscosity [6].

In a clinical setting, measurements cannot always be performed in a standardized manner as patients may not be able to adhere to breathing instructions, may have eaten prior to the examination, or may have postoperative changes (for example, resected liver lobes or surgical dressings), which may make only the left lobe of the liver accessible for examination. A comprehensive understanding of measured values and the factors influencing clinical liver measurements (SWS, SWD, ATI) is of great clinical importance because it helps clinicians evaluate whether values taken in a state that differs from the recommended one are trustworthy. As these values contribute to clinical staging, they also have a direct implication on patient management, with inadequate values leading to incorrect treatment pathways. A better insight into the influencing factors of multiparametric liver imaging also provides a basis for a standardized protocol and, hence, more accurate and reliable staging. To date, research into the breathing phase, prandial state, and liver lobe for SWS measurements, and to a certain extent, for ATI measurements, has already been established [16,17,18,19,20,21,22,23,24,25]. No such research exists for SWD, but it is equally important given the increasing use of SWD in clinical settings. The goal of this research is, therefore, to investigate the effect of breathing phase (expiration versus inspiration), liver lobe (right versus left), and prandial state (fasting versus nonfasting) on SWS, SWD, and ATI measurements.

## 2. Materials and Methods

### 2.1. Study Design

This study was approved by the local ethics committee, and all volunteers gave informed consent. From November to December 2022, 20 healthy volunteers (11 male, 9 female) were recruited (Table 1). None of the volunteers had a history of cancer, hepatitis, or diabetes. One participant reported alcohol usage greater than >140 g per week (average safe drinking guidelines for European nations vary between 10–40 g per day [26]). SWS, SWD, and ATI measurements were obtained in several positions, and measurement details are summarized in Table 2. Measurements were first taken in the recommended position for SWE: in the right lobe, on a breath-hold following gentle expiration, and with the volunteer in a fasting state [4]. This reference position is referred to as Right Expiration Fasting (REF). Three test conditions were then assessed:aFollowing inspiration (Right Inspiration Fasting, RIF);bIn the left lobe (Left Expiration Fasting, LEF);cWith volunteers in a nonfasting state (Right Expiration Nonfasting, RENF).

Prior to fasting-state measurements, volunteers fasted for a minimum of 3 h. Measurements taken in the right lobe were completed using an intercostal approach, with the subject supine, relaxed, and with arm raised above the head. Measurements in the left lobe were obtained via an abdominal approach, with the probe sagittally oriented, approximately in line with the abdominal aorta. For nonfasting measurements, volunteers consumed a calorie-controlled meal of approximately 450 ± 25 kilocalories with a breakdown of approximately 55% carbohydrate, 25% fat, and 20% protein. Participants were asked to eat the meal within 10 min, and measurements were taken 30 min thereafter. All measurements were acquired by a single, primary operator (for comparison of measurement positions). In addition, a secondary operator also acquired measurements in the REF position for an interoperator reliability assessment. Both operators are experienced examiners with >5 years’ experience in radiology.

### 2.2. Equipment and Protocol

Measurements were performed on a Canon Aplio i800 with the Canon i8CX1 convex array (centre frequency 4 MHz, bandwidth 1–8 MHz), using the default Multishot SWE and ATI (i.e., no custom changes to the default machine protocols for SWE and ATI were made). For each acquisition, after starting the Multishot mode, the operator waited until wavefronts in the propagation map appeared as stable, as smooth, and as parallel as possible, before freezing the image and taking measurements. The SWE ‘Acquisition Box’ was placed 10 mm below the liver capsule, and a single circular Region of Interest (ROI) 10 mm in diameter was placed in the region of the acquisition box where the lines appeared the most smooth and parallel as possible (Figure 1). Previous studies in phantoms showed that the most stable measurement area was at the top of the acquisition box on the side closest to the push pulse [27]. However, in humans, vessels and other inhomogeneities in the liver meant that the ROI was placed freely within the acquisition box (Figure 1b). In each position, nine measurements were obtained for SWS and SWD. In this paper, ‘stiffness’ values are reported in terms of speed. For ATI (Figure 1c,d), the ‘ATI Box’ size was set to a maximum and placed below the liver capsule (Figure 1d). Within the ATI box, the ROI size was adjusted to approximately 3–4 cm axially and 2–3 cm in the transverse direction. In accordance with the manufacturer’s guidelines, it was placed just below the orange band at the top of the ATI box (indicative of the liver capsule artefact), avoiding large vessels and dark-blue (deep) areas. In all ATI measurements, the profile uniformity index (R² value, Figure 1c) was greater than 0.90, and five measurements were taken in each position. Measurements in the left side of the liver were not possible due to the limited depth of the left liver lobe visible from an abdominal approach.

### 2.3. Data Processing and Statistics

For SWS, measurements with an Interquartile Range (IQR) over Median ratio (IQR/M) less than 0.15 are deemed reliable by the World Federation for Ultrasound in Medicine and Biology Guidelines [4]. No quality metric exists for SWD at present; however, since the dispersion technique is predicated on a good SWS measurement, an IQR/M < 0.15 for SWS is considered as a useful starting point for obtaining reliable SWD measurements. Statistical analyses were completed using Matlab R2021a (The Mathworks, Inc., Natick, MA, USA). SWS, SWD, and ATI values were taken as the median value of the set of measurements [4], as displayed on the multiparametric report on the Canon system. Multiple *t*-tests were used to compare test conditions to the reference (REF) condition. Normality was evaluated using Shapiro–Wilk tests, and equal variance was assessed by Bartlett’s tests. Where the data were normally distributed and variances were equal, a Student’s *t*-test was used. Where data were not normally distributed, a Wilcoxon rank test was used, and where variances were not equal, the Welch’s *t*-test was used. Pearson’s Correlation Coefficient (PCC) was used to assess correlation. To assess interobserver reliability, a two-way, random-effects, single-rater, absolute agreement Intraclass Correlation Coefficient (ICC) test was used [28]. Bland–Altman plots were also produced to assess and graphically illustrate interobserver variability, and 95% limits of agreement were calculated. Finally, to assess agreement on clinical staging, data were plotted in a scatter plot with clinical staging cutoffs as displayed in Table 3.

## 3. Results

### 3.1. Data Quality

In B-Mode, no signs of hepatic damage or fibrosis were visible—the liver surface appeared regular, the parenchyma homogeneous, and no venous irregularities were seen. The IQR/M was less than 0.15 for all SWS measurements in all volunteers in all positions. The IQR/M was less than 0.30 for all SWD measurements in all volunteers in all positions. The quality metric for ATI imaging was greater than, or equal to, 0.90 for all measurements in all volunteers in all positions.

### 3.2. SWS, SWD, and ATI Results

A full result summary is given in Table 4. For SWS (Figure 2), no significant differences were observed. For SWD (Figure 3), the mean of measurements in the left lobe (12.18 m/s/kHz) was significantly higher than the reference (10.81 m/s/kHz, * *p* = 0.021). SWS and SWD measurements were strongly correlated across all conditions (PCC, r = 0.805, *p* < 0.001). For ATI (Figure 4), no significant differences existed between any test condition and the reference condition, and ATI measurements were not significantly correlated with either SWS (PCC, r = 0.097, *p* = 0.39) or SWD (PCC, r = −0.002, *p* = 0.98).

### 3.3. Reliability

ICC values (Table 5) show moderate to good interobserver reliability for almost all measurements. Bland–Altman analysis (Figure 5) with one-way *t*-test indicates the difference values between operators were not significantly different than the expected zero mean for SWS, SWD, and ATI (*p* = 0.584, *p* = 0.185, and *p* = 0.766, respectively). Operators’ clinical staging outcome matched in 100% of volunteers for SWS and ATI (Figure 6 left and right panels), whereas it matched in 80% of volunteers for SWD (Figure 6 middle panel).

### 3.4. Variability

Figure 7 and Figure 8 and Table 6 present the results for individual measurement variability. On the Canon system, the measurement (average of values in the ROI) as well as the standard deviation of the measurement are displayed (as seen on the top left of Figure 1A). A high standard deviation as a proportion of the measurement value indicated significant variability of the measured region. In order to assess, whether certain conditions and positions have higher variability, the Coefficient of Variance (CoV, calculated as the standard deviation divided by the mean) was calculated for all measurements, and the results are displayed in Figure 7 and Figure 8. The average CoV is presented in Table 6 and demonstrates that, for both SWS and SWD, the most individual measurement variation is seen in the left lobe.

## 4. Discussion

This study investigating the effect of different measurement conditions found no significant influence of breathing and prandial state on SWS, SWD, and ATI values. SWS and SWD measurements were strongly correlated. SWD measurements in the left lobe showed a higher individual measurement variability. Interobserver agreement was moderate to good for SWS, SWD, and ATI.

### 4.1. Data Quality

It has previously been suggested that hotspots should be avoided for SWD where possible and that the IQR/M < 0.15 quality metric for SWS be used as an indication of data quality for SWD measurements as well [15]. In this study, all SWS measurements had an IQR/M < 0.15. A quality metric for SWD has not yet been defined; however, in this study, all SWD measurements had an IQR/M < 0.30. This may provide an indication of the expected quality for SWD measurements in healthy volunteers when SWS meets the published criteria. However, it should be noted that there is no current guideline stating the target quality for SWD. Finally, for ATI, images were only acquired when the quality metric (R^2^) showed a value of R^2^ > 0.90, as per manufacturer guidelines.

### 4.2. SWS

Inspiration: There was no significant difference between SWS measurements on inspiration compared to the reference position. Previous studies also found that there was no difference between inspiration and expiration [18], but other studies suggest that stiffness may increase during inspiration [19] and yet others [20,24] reported lower stiffness after inspiration than after expiration. It is hypothesized that changes to SWS may be due to changes in intrathoracic pressure and hepatic venous return, which occur during respiration [24]. However, given the disparity in the literature, it is likely that the differences are not purely physiological but are also influenced by the method of elastography, machine parameters, and operator acquisition methods. Variability in measurements due to deep inspiration has previously been noted and discussed in the European Federation of Societies for Ultrasound in Medicine and Biology guidelines [29].

Left Lobe: No significant difference between LEF and REF was found. However, in their 2011 study, Karlas et al. [19] reported significantly higher values in the left lobe when studying 12 healthy volunteers using a Siemens Acuson S2000 system (Siemens Medical Solutions, Mountain View, CA, USA). It is, therefore, possible that differences exist when the measurements are performed on different machines, all of which utilize slightly different shear-wave-generation and acquisition protocols.

Nonfasting: There is no significant difference between SWS in the RENF and REF conditions. Previous results published by Silva et al. [25] also show no difference in 22 healthy volunteers 30 min postprandially, when using TE. With ARFI, Kaminuma et al. [18] also observed no difference in 20 healthy volunteers. However, both Mederacke et al. [21] and Popescu et al. [30] showed an increase in liver stiffness values up to (and including) one hour postprandial using TE and ARFI, respectively. Other literature shows the maximum increase in portal blood flow is obtained 30 min postprandially [31,32], but, in our study, we did not find any significant differences at this timepoint. Another explanation for discrepancies in literature may be due to changes in meal composition and total calories, but further work specifically on meal ingestion would be required for a definite conclusion.

### 4.3. SWD

Inspiration: There is a strong correlation between SWS and SWD measurements, which matches the correlation between viscosity and elasticity in the liver previously shown by Chen et al. [33]. Thus, in the RIF condition, no significant difference in SWD values is expected, given that SWS also showed no significant difference in RIF compared to REF.

Left Lobe: However, surprisingly, LEF SWD measurements were significantly higher than REF measurements, even though that was not the case for SWS. This indicates that SWD is likely more sensitive to slight variations from motion or probe placement artefacts compared to SWS. This can be the case, given that SWD measurements rely on individual speed estimates (known as phase velocities) taken throughout the entire frequency range in order to calculate a ‘slope’ of the change. However, SWS measures the group velocity (a type of overall average velocity across all frequencies together). Therefore, inaccuracies in shear wave measurements at higher frequencies (for example, caused by signal loss due to attenuation) could easily lead to a change in SWD slope estimate, but may not significantly affect the group velocity (SWS). Specifically, in the left lobe, cardiac motion artefacts are more pronounced than in the right for SWS measurements [34]. There is also greater freedom when placing the probe abdominally and, thus, a higher probability of tilting the probe, which may lead to a decrease in the intensity of the ‘push pulse’ [27].

Furthermore, in the left lobe, the CoV for SWD was the highest of the CoVs for all measurements. This indicates that individual measurements are more variable and contain higher uncertainties than measurements in other locations. Therefore, for general liver assessment, SWD measurements should not be taken in the left lobe, given the increased range of values in this area and the high standard deviation of these measurements.

Nonfasting: No differences were observed between fasting and nonfasting measurements. However, measurements were only performed at a single timepoint. Although this timepoint was selected to be at the maximum increase in portal blood flow, no difference was observed, even though some previous studies did observe SWS differences [21,30]. As discussed for SWS, it is possible that meal composition may have played a role in increasing or decreasing the measurement values.

Overall: SWD values in this study appeared to be higher than values previously reported in Sugimoto et al. [14]. However, the cutoffs present in the Canon machine appear to be more reasonable for the present, healthy cohort. Overall, the cutoff values for SWD are still under investigation: cutoffs were initially determined using biopsy samples from a patient cohort recruited exclusively with suspected NAFLD [14], but a cohort with a different aetiology could have different cutoff values, as suggested by the publication of Lee et al. [35], and thus more varied aetiologies may be more representative. Furthermore, within the published cutoff values, there remains a high degree of overlap between different inflammation stages [14]. Therefore, the use of the new manufacturer cutoff values seems to be more in line with what is expected from a healthy European cohort.

### 4.4. ATI

No difference was observed between ATI RENF and REF, which matches results published by Silva et al. [25], who also show no change in CAP in volunteers in fasting and nonfasting states. Evaluation of ATI for inspiration versus expiration has not yet been reported in literature, and in this study, no significant differences were found.

### 4.5. Reliability

This study demonstrated good and moderate interobserver reliability for SWS, SWD, and ATI. SWS values presented in this study are in line with those reported by Kishimoto et al. [36], who showed an interoperator ICC of 0.79 at 4 cm depth in the right lobe of the liver. However, it should still be noted that the use of the ICC statistic is limited in these cases, given that it is typically heavily dependent on the range of the data being measured, with a wider range of data leading to better ICC results [37]. Thus, due to the low range of data obtained from only healthy volunteers, the true ICC value across a population of both healthy and unhealthy livers may yield much higher ICC agreement values. When looking at clinical staging, good interoperator agreement and primary operator repeatability was seen for SWS and ATI. For dispersion, however, only 80% of participants were classified into the same clinical category. As previously discussed, this may be a result of using cutoff values that are still under investigation, as well as a lack of standalone SWD quality metrics. Furthermore, in [27], SWD is shown to be two to three times more variable than SWS, even in homogeneous phantoms. It is, therefore, recommended that the SWD measurement is only used as part of a multiparametric liver assessment until further investigation into the cutoff values and the quality metric has been completed.

### 4.6. Limitations

A limitation of this work is that the study only contained a healthy European adult cohort, but results may differ in paediatric populations or in a diseased liver patient cohort. In addition, measurements taken postprandially were only taken at a single timepoint, and different meal compositions were not tested.

## 5. Conclusions

For the first time, this study investigates the differences in the new ultrasound measurement, SWD, in relation to liver lobe, breathing state, and prandial state. Breathing and prandial state did not significantly affect SWS, SWD, and ATI values. SWS and SWD measurements were strongly correlated. SWD measurements in the left lobe showed a higher individual measurement variability and should, therefore, be avoided. Interobserver agreement was moderate to good for SWS, SWD, and ATI.

## Figures and Tables

**Figure 1 diagnostics-13-00989-f001:**
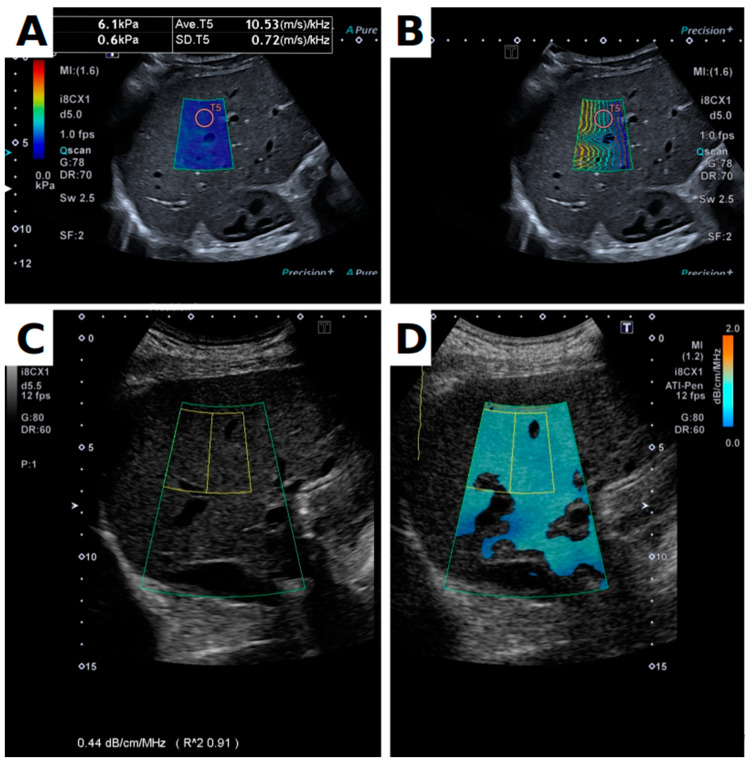
Panel (**A**) shows a B-mode image with spatial distribution of shear wave speed overlay. The Shear Wave (SW) box displays the SW map of the region. The Region of Interest (ROI) is indicated by the red circle located in a homogeneous region of the map. Note also that the ROI is located at a region with smooth and parallel lines in the shear wave propagation map, shown in panel (**B**). Panels (**C**,**D**) show the Attenuation Imaging (ATI) display. In panel (**C**), only the B-mode image is displayed, and the quality metric (R²) is displayed at the bottom of the image. In (**D**), the ATI overlay with ATI box (large region) and ATI ROI (smaller box with yellow outline) are visible. ATI ROIs were placed directly below the orange liver capsule artefact, in the light-blue region, as per manufacturer’s guidelines.

**Figure 2 diagnostics-13-00989-f002:**
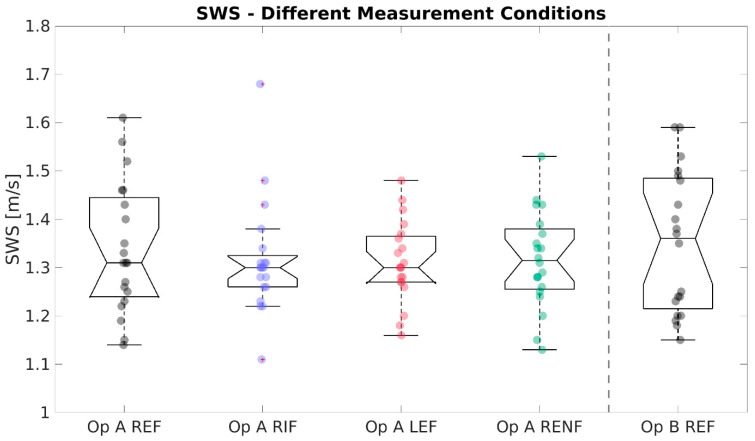
Boxplots of Shear Wave Speed (SWS) measurements (m/s) for the different conditions (standard Right Expiration Fasting (REF), Right Inspiration Fasting (RIF), Left Expiration Fasting (LEF), and Right Expiration Nonfasting (RENF)) performed by operator A (Op A) and REF performed by operator B (Op B). Each circle represents the median of nine repeated measurements in one individual. No significant differences in mean existed between the reference condition “Op A REF” and any other condition.

**Figure 3 diagnostics-13-00989-f003:**
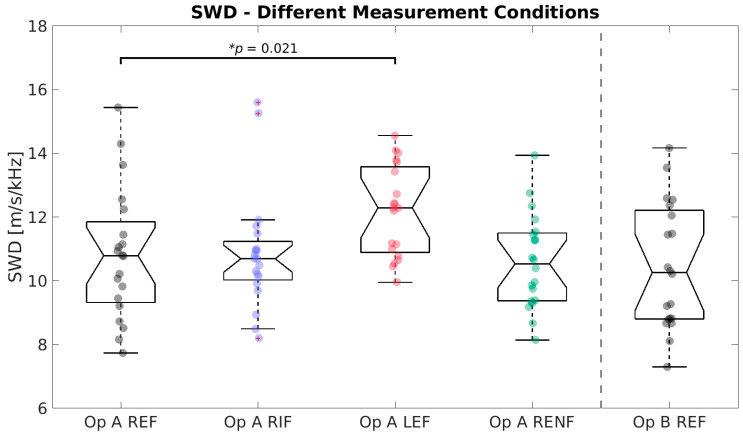
Boxplots of Shear Wave Dispersion (SWD) measurements for the different conditions (standard Right Expiration Fasting (REF), Right Inspiration Fasting (RIF), Left Expiration Fasting (LEF), and Right Expiration Nonfasting (RENF)) performed by operator A (Op A) and REF performed by operator B (Op B). Each circle represents the median of nine repeated measurements in one individual. * indicates a statistical significant difference between a test condition with the reference condition “Op A REF”.

**Figure 4 diagnostics-13-00989-f004:**
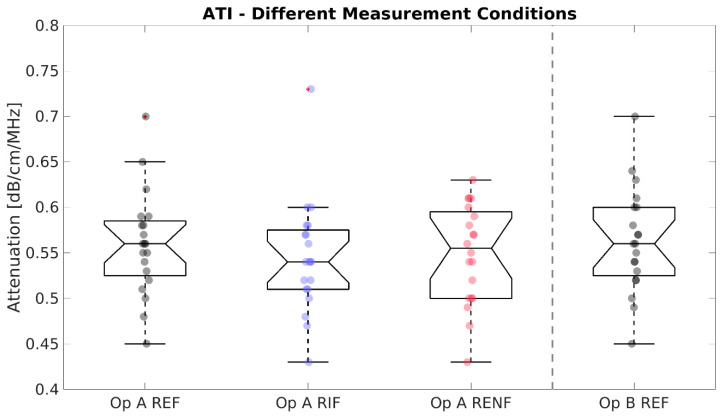
Attenuation Imaging (ATI) measurements for the different conditions (standard Right Expiration Fasting (REF), Right Inspiration Fasting (RIF), and Right Expiration Nonfasting (RENF)) performed by operator A (Op A) and REF performed by operator B (Op B). Each circle represents the median of five repeated measurements in one individual. No significant differences in mean existed between the reference condition “Op A REF” and any other condition.

**Figure 5 diagnostics-13-00989-f005:**
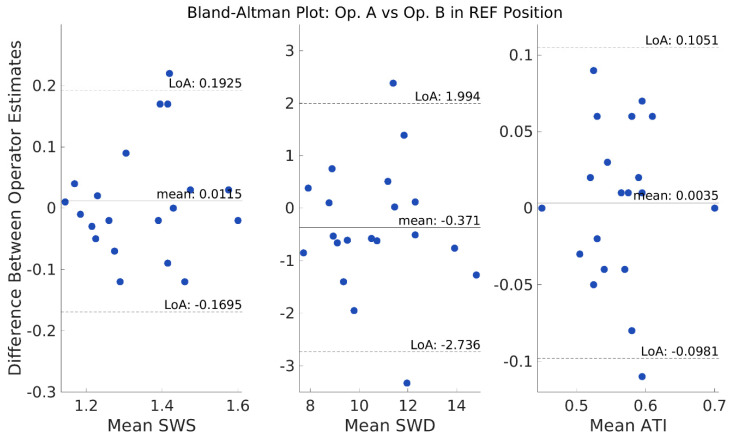
Interoperator Bland–Altman plots for SWS, SWD, and ATI values in the reference position Right Expiration Fasting (REF) with the x-axis indicating the mean and the y-axis indicating the difference between both operator’s median of nine (for SWS and SWD) or five (for ATI) repeated measurements for each individual (blue circle). Limits of Agreement (LoA) calculated as 1.96 times the standard deviation (SD) are displayed as dashed lines on the plots. Op = operator.

**Figure 6 diagnostics-13-00989-f006:**
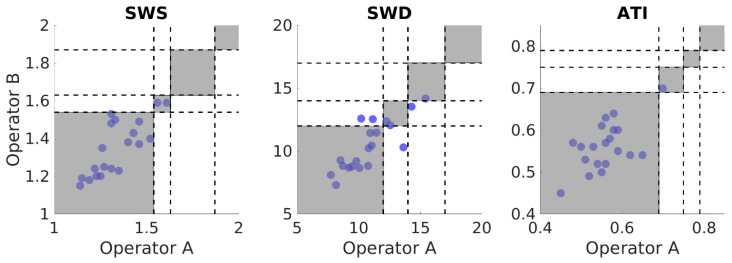
Scatter plots of primary (Operator A) versus secondary (Operator B) results for Shear Wave Speed (SWS), Shear Wave Dispersion (SWD), and Attenuation Imaging (ATI) in the reference condition Right Expiration Fasting (REF). Each circle represents the median of nine (for SWS or SWD) or five (for ATI) repeated measurements in one individual. Black, dashed lines denote cutoff values for different disease stages for SWS, SWD, and ATI. Grey areas indicate agreement in clinical staging between operators; white areas indicate disagreement.

**Figure 7 diagnostics-13-00989-f007:**
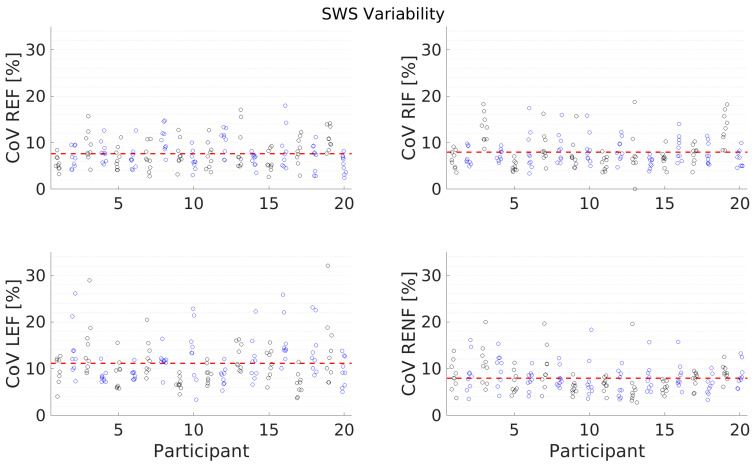
Coefficient of Variation (CoV), calculated as the standard deviation divided by the mean for every individual SWS measurement point (circles) for all participants and all conditions (Right Expiration Fasting (REF), Right Inspiration Fasting (RIF), Left Expiration Fasting (LEF), and Right Expiration Nonfasting (RENF)). The red, dashed line indicates the average CoV of all measurements.

**Figure 8 diagnostics-13-00989-f008:**
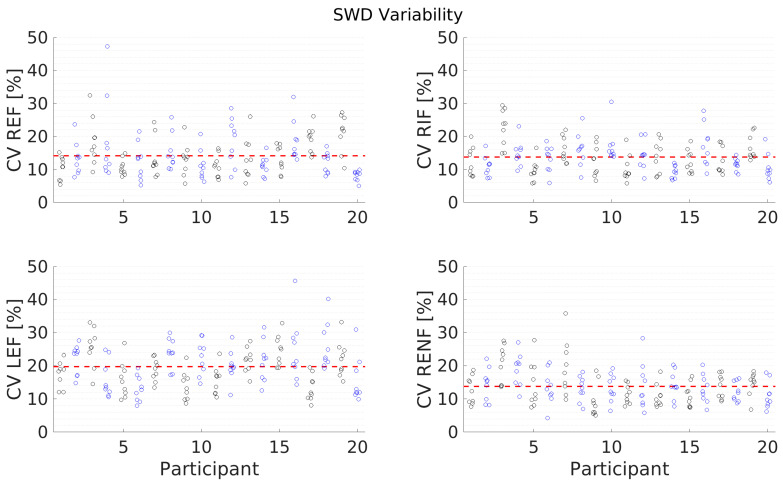
Coefficient of Variation (CV), calculated as the standard deviation divided by the mean for every individual SWD measurement point (circles) for all participants and all conditions (Right Expiration Fasting (REF), Right Inspiration Fasting (RIF), Left Expiration Fasting (LEF), Right Expiration Nonfasting (RENF)). The red, dashed line indicates the average CV of all measurements.

**Table 1 diagnostics-13-00989-t001:** Summary of volunteer physical parameters. *n* = 20 adult volunteers (9 female, 11 male).

	Mean ± Standard Deviation	Minimum	Maximum
Age (Years)	31.8 ± 5.3	24.0	45.0
Weight (kg)	73.8 ± 12.4	51.0	97.0
Height (cm)	178.1 ± 9.9	158.0	194.0
Body Mass Index (kg/m^2^)	23.1 ± 2.6	18.4	27.7

**Table 2 diagnostics-13-00989-t002:** Summary and description of the test conditions evaluated against the reference condition, which is the current recommended approach to SWE measurements. IC = intercostal. Abb. = abbreviation in manuscript.

Full Name	Abb.	Measurement Purpose	Test Condition (vs. Reference)	Measurement Description
Right Expiration Fasting	REF	Reference/ Standard Measurement Condition	N/A	3 h fast, IC approach, subject supine, relaxed, breath hold following gentle expiration
Right Inspiration Fasting	RIF	Test Condition	Inspiration	3 h fast, IC approach, subject supine, relaxed, breath hold following gentle inspiration
Left Expiration Fasting	LEF	Test Condition	Left Lobe	3 h fast, abdominal approach, subject supine, relaxed, breath hold following gentle expiration
Right Expiration Nonfasting	RENF	Test Condition	Nonfasting	0.5 h postprandial, IC approach, subject supine, relaxed, breath hold following gentle expiration

**Table 3 diagnostics-13-00989-t003:** Categorization of fibrosis, inflammation, and steatosis based on SWS, SWD, and ATI measurements using the Canon Aplio i800 machine. Cat. = category. Cutoff values for SWS, SWD, and ATI are based on manufacturer recommendations. Shading is used to highlight values associated with corresponding categories.

Technique	Cat. 1	Value	Cat. 2	Value	Cat. 3	Value	Cat. 4	Value
SWS (m/s)	F0/F1	<1.54	F2	1.54–1.63	F3	1.64–1.87	F4	>1.87
SWD (m/s/kHz)	A0	<12.0	A1	12.0–14.0	A2	14.1–17.0	A3	>17
ATI (dB/cm/MHz)	S0	<0.69	S1	0.69–0.75	S2	0.76–0.79	S3	>0.79

**Table 4 diagnostics-13-00989-t004:** Summary of mean and standard deviation and statistical test results between the reference and test conditions. Units of measurement for mean ± standard deviation (SD) are [m/s] for SWS, [m/s/kHz] for SWD, and [dB/cm/MHz] for ATI. * indicates statistical significance (*p* < 0.05). S-W stands for Shapiro Wilk.

	Mean ± SD	Test Condition	Mean ± SD	Equal Var. (Bartlett’s)	Normality (S-W test)	*t*-Test Used	Final *p*-Value
SWS REF	1.34 ± 0.13 m/s	SWS RIF	1.32 ± 0.12	Y (*p* = 0.532)	N (* *p* = 0.002)	Wilcoxon	0.551
		SWS LEF	1.31 ± 0.08	N (* *p* = 0.043)	Y (*p* = 0.869)	Welch’s	0.465
		SWS RENF	1.32 ± 0.10	Y (*p* = 0.211)	Y (*p* = 0.989)	Student’s	0.403
		Op B SWS REF	1.35 ± 0.15	Y (*p* = 0.714)	Y (*p* = 0.077)	Student’s	0.584
SWD REF	10.81 ± 2.05 m/s/kHz	SWD RIF	10.87 ± 1.84	Y (*p* = 0.639)	N (* *p* = 0.005)	Wilcoxon	0.903
		SWD LEF	12.18 ± 1.41	Y (*p* = 0.115)	Y (*p* = 0.216)	Student’s	* 0.021
		SWD RENF	10.59 ± 1.48	Y (*p* = 0.162)	Y (*p* = 0.858)	Student’s	0.596
		Op B SWD REF	10.44 ± 1.97	Y (*p* = 0.858)	Y (*p* = 0.277)	Student’s	0.185
ATI REF	0.56 ± 0.06 dB/cm/MHz	ATI RIF	0.54 ± 0.06	Y (*p* = 0.415)	N (* *p* = 0.035)	Wilcoxon	0.343
		ATI RENF	0.55 ± 0.05	Y (*p* = 0.837)	Y (*p* = 0.521)	Student’s	0.355
		Op B ATI REF	0.56 ± 0.06	Y (*p* = 0.975)	Y (*p* = 0.822)	Student’s	0.766

**Table 5 diagnostics-13-00989-t005:** Summary of inter-operator reliability results using the two-way, random-effects, single-rater absolute agreement Intraclass Correlation Coefficient (ICC) and Bland–Altman analysis. The two operators used were the primary operator A and a secondary operator B. Rating scale for ICC: <0.5, poor; 0.5–0.75, moderate; 0.75–0.9, good; >0.9, excellent. B-A = Bland–Altman. LoA = limits of agreement, calculated at 1.96 times the standard deviation on Bland–Altman plots.

Metric	ICC	ICC 95% CI	ICC Rating	B-A Mean Diff.	B-A LoA
SWS	0.79	[0.54, 0.91]	Good	0.01 m/s	[−0.17, 0.19] m/s
SWD	0.81	[0.59, 0.92]	Good	−0.37 m/s/kHz	[−2.74, 1.99] m/s/kHz
ATI	0.60	[0.22, 0.82]	Moderate	0.00 dB/cm/MHz	[−0.10, 0.11] dB/cm/MHz

**Table 6 diagnostics-13-00989-t006:** Coefficient of Variance (CoV) for measurements—variability of individual measurements of operator A (Right Expiration Fasting (REF), Right Inspiration Fasting (RIF), Left Expiration Fasting (LEF), and Right Expiration Nonfasting (RENF)) and operator B (Op B) Right Expiration Fasting (REF).

Metric	REF	RIF	LEF	RENF	Op B REF
CoV SWS (%)	7.63	7.94	11.08	7.95	6.90
CoV SWD (%)	14.16	13.74	19.68	13.77	12.07

## Data Availability

Data can be made available on reasonable request as per the institute’s internal data governance policy.

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
