# Peer review of "Impact of Breathing Phase, Liver Segment, and Prandial State on Ultrasound Shear Wave Speed, Shear Wave Dispersion, and Attenuation Imaging of the Liver in Healthy Volunteers"

_diagnostics, 2023, doi:10.3390/diagnostics13050989_

Round 1
Reviewer 1 Report (Previous Reviewer 1)
I'd like to thank to Authors for the thorough revision of manuscript that is now greatly improved.
Author Response
Thank you for your positive feedback. We have now read the manuscript again and corrected minor spelling errors.

Reviewer 2 Report (Previous Reviewer 3)
This manuscript is the revised version. The authors have revised the manuscript extensively. The presentations of the Materials and Methods as well as Results are sufficiently clear and detailed. The Discussion is also well-written. The reviewer has just one last suggestion:
The purpose of this study is to investigate how breathing phase, liver segment, and prandial state influence the SWS, SWD and ATI. However, in the Introduction section, the authors did not describe what motivates them to investigate this topic. In other words, the motivation of this study is not clear. In addition, the authors should describe the clinical relevance of understanding how breathing phase, liver segment, and prandial state influence the SWS, SWD and ATI. The reviewer suggests to improve these two points, and then the manuscript can be recommended to be accepted for publication.
Author Response
Dear Reviewer
We thank you for your positive feedback.
We have now revised the introduction, expanding the section focusing on the clinical motivation and relevance. We have also revised the manuscript regarding spelling errors.
We hope the changes we made are in line with your expectations.
Kind regards
The authors
Changes:
"In a clinical setting, measurements cannot always be performed in a standardized manner as patients may not be able to adhere to breathing instructions, may have eaten prior to the examination, and may have postoperative changes (for example, resected liver lobes or surgical dressings), which may make only the left lobe of the liver accessible for examination. A comprehensive understanding of measured values and the factors influencing clinical liver measurements (SWS, SWD, ATI) is of great clinical importance because it helps clinicians evaluate whether values taken in a state that differs from the recommended one are trustworthy. As these values contribute to clinical staging, they also have a direct implication on patient management, with inadequate values leading to incorrect treatment pathways. A better insight into the influencing factors of multiparametric liver imaging also provides a basis for a standardized protocol and hence more accurate and reliable staging. To date, research into the breathing phase, prandial state, and liver lobe for SWS measurements, and to a certain extent, for ATI measurements, has already been established [16-25]. No such research exists for SWD, but it is equally important given the increasing use of SWD in clinical settings. The goal of this research is therefore to investigate the effect of breathing phase (expiration versus inspiration), liver lobe (right versus left), and prandial state (fasting versus non-fasting) on SWS, SWD, and ATI measurements."

This manuscript is a resubmission of an earlier submission. The following is a list of the peer review reports and author responses from that submission.
Round 1
Reviewer 1 Report
The aim of this study was to assess whether the breathing phase (inspiration vs expiration), meal ingestion and taking measurement in different liver lobes may affect the value of shear wave speed, shear wave dispersion and attenuation coefficient. A Canon ultrasound system was used. The study could be of interest, however there are some issues that limit the value of the findings.
These are my comments:
- There is a methodological mistake that is unacceptable: to maintain the IQR/M in the range recommended by guidelines, measurements that were “unusually high” were deleted (deselecting them in the final report) after having accepted them during the examination. It’s unclear how many measurements were deleted and what was the “standard” for considering the measurement “unusually high”. If a measurement is judged not good, it should not be recorded in the first place, and once accepted and recorded must not be deleted just to comply with the guideline’s recommendations. Measurements with an IQR/M higher than 0.15 (for values given in m/s) must be considered unreliable and not artificially converted to “reliable”. On this regard, the reviewer would like to highlight that this is not a study conducted in “real-life” conditions but a study planned in healthy volunteer. Therefore, it doesn’t make much sense to state that “During a clinical exam (which is time-limited), several measurements using multiple images and ROIs are captured by the clinician”.
-Since this study was designed to specifically assess the influence of meal ingestion on the SWS, SWD and ATI measurements it is unclear why measurements were not performed at 30 minutes after eating, i.e., at the time of the maximum increase in portal blood flow. Please explain.
-The use of the Gwet’s agreement coefficient is quite weird and doesn’t make any sense since this is a series of healthy volunteers, therefore it’s likely that they all belong to the same clinical stage. Please explain.
-The study is underpowered and this might have affected the statistical significance of the results. On this regard, it’s unclear why the power of the study was not assessed in advance.
-It’s incorrect to state that ATI measurements in the right and left lobe were the same because measurements in the left lobe were not performed at all due to the limited depth of the left lobe as stated in lines 125-126.
-Several studies have shown a good to excellent intra- and inter-observer agreement in SWS assessment, and the same as been reported for ATI. Please compare your data to the literature and give some insights for this discordance.
-Lines 43-44: This statement is somehow incorrect. CAP is not derived from a TE measurement and is not related to it. CAP is obtained analyzing the attenuation of the US beam that is used to track the velocity of the shear wave, thus they are unrelated to each other. By the way, it’s incorrect to state that it is obtained in a “cylindrical region”: this statement pertains to TE not to CAP.
-Lines 45-46: This statement is incorrect. Attenuation coefficient is always a “single value” even with ATI algorithms. Please modify.
-Line 70: this statement is incorrect and the self-citation is inappropriate because the cited study assessed sources of variability in shear-wave speed and dispersion in a phantom, therefore has nothing to do with alcohol intake! By the way, >80 g alcohol/day is already a very harmful alcohol consumption and identifies heavy drinkers. There is variability in the cut-off value used to determine the significance of current alcohol consumption. The range of current alcohol intake that has been proposed for this cut-off is 10 g per day or less to 20–40 g per day in men and 20 g per day in women. It has been suggested that a reasonable cut-off would be 20 g per day [Kwon HK, et al. Effect of lifetime alcohol consumption on the histological severity of non-alcoholic fatty liver disease. Liver Int 2014; 34:129-35].
-Line 111: the shear modulus is used in MRE not in US SWE elastography. Please correct.
-Lines 114-114: It’s the opposite: the Young's modulus is related to shear wave speed and not vice versa.
Reviewer 2 Report
This article is very innovative, with three different test methods to detect the same people, which can provide a reference for ultrasound clinics. However, there are several question needed to pay attention:
1.Please clarify the purpose of this study, is it to compare the merits of these three methods?
2.Pleas to discuss the difference of this study compared to the previous studies.
3.For the conclusion part, I hope the author could be concise and accurate.
4. Where available, we recommend expanding the sample size to make the results more credible and convincing.
Reviewer 3 Report
The manuscript is well-written and is written in great detail. The reviewer has no any problems regarding the study design and writing of the manuscript. It is an excellent work.
The only suggestion for the authors is that, please add at least one paragraph to the Introduction section to talk about the clinical background regarding liver diseases of this study. Currently, there is only one sentence about the clinical background, and the rest of the Introduction section is all about the technical background (regarding SWE, SWD and ATI). The statement of the clinical background is also important.
In addition, please add some sentences to the Introduction section to clearly describe the clinical motivation that motivates you to do this study. It is important to let readers with clinical background to understand the clinical motivation of this study such that they will know how to apply the technique and knowledge in this study to clinical applications.